# The Interplay of Type 1, Type 2, and Type 3 Lymphocytes and Cytokines in Atopic Dermatitis

**DOI:** 10.3390/ijms24043310

**Published:** 2023-02-07

**Authors:** Keiichi Yamanaka, Yui Kono, Shohei Iida, Takehisa Nakanishi, Mai Nishimura, Yoshiaki Matsushima, Makoto Kondo, Koji Habe, Yasutomo Imai

**Affiliations:** 1Department of Dermatology, Mie University Graduate School of Medicine, 2-174 Edobashi, Tsu 514-8507, Japan; 2Imai Adult and Pediatric Dermatology Clinic, 5-1-1 Ebie, Fukushima, Osaka 553-0001, Japan

**Keywords:** atopic dermatitis, inflammatory skin mouse model, cytokine, type 2 inflammation, interleukin-17E, interleukin-25

## Abstract

Atopic dermatitis (AD) is classified as a type 2 disease owing to the majority of type 2 lymphocytes that constitute the skin-infiltrating leukocytes. However, all of the type 1–3 lymphocytes intermingle in inflamed skin lesions. Here, using an AD mouse model where caspase-1 was specifically amplified under keratin-14 induction, we analyzed the sequential changes in type 1–3 inflammatory cytokines in lymphocytes purified from the cervical lymph nodes. Cells were cultured and stained for CD4, CD8, and γδTCR, followed by intracellular cytokines. Cytokine production in innate lymphocyte cells (ILCs) and the protein expression of type 2 cytokine IL-17E (IL-25) were investigated. We observed that, as inflammation progresses, the cytokine-producing T cells increased and abundant IL-13 but low levels of IL-4 are produced in CD4-positive T cells and ILCs. TNF-α and IFN-γ levels increased continuously. The total number of T cells and ILCs peaked at 4 months and decreased in the chronic phase. In addition, IL-25 may be simultaneously produced by IL-17F-producing cells. IL-25-producing cells increased in a time-dependent manner during the chronic phase and may work specifically for the prolongation of type 2 inflammation. Altogether, these findings suggest that inhibition of IL-25 may be a potential target in the treatment of inflammation.

## 1. Introduction

Skin inflammation is not just a localized condition of the skin, but also causes inflammation in other organs throughout the body, leading to comorbidities and a shortened life span. Among them, atopic dermatitis (AD) is a chronic intractable form of dermatitis. Type 2 cytokines, such as IL-4 and IL-13, produced by T helper (Th)-2 cells and innate lymphocyte cells (ILC)-2, are involved in the pathogenesis of AD [1,2]. In the acute phase of AD, IL-4 produced by ILC2 and Th2 induces pruritus, and IL-13 decreases the production of antimicrobial peptides and filaggrin, destroying the skin barrier. IL-31, produced by Th2 cells, also leads to similar outcomes. IL-33 and IL-18 are proinflammatory cytokines that play pivotal roles in allergic disorders. In transgenic mice expressing IL-33 or IL-18 driven by a keratin-14 promoter (IL33tg and IL-18tg), AD-like inflammation develops spontaneously with the activation of ILC2s [2,3], contributing to the development and exacerbation of type 2 immunity. In the chronic phase, type 2 immunity plays a significant role; however, in addition to type 2 lymphocytes, type 1 cells producing TNFα and IFNγ along with type 3 cells producing IL-17A and IL-17F are thought to coexist in lesions at the site of skin inflammation. IL-22 promotes keratinocyte proliferation [4,5,6]. A previous study demonstrated that Th22 and Th17 immune responses contribute to chronic skin lesions in AD, especially in pediatric, intrinsic, and Asian patients [7]. To date, there have been no studies on temporal and spatial changes in the number of cytokine-producing cells. This study examined the quantitative and qualitative dynamics of type 1, type 2, and type 3 inflammations over time.

IL-17E (IL-25) is a novel IL-17 cytokine family produced by Th2 cells, mast cells, and epithelial cells. IL-25 induces eosinophilia and IgE production by producing Th2 cytokines such as IL-4, IL-5, and IL-13. IL-25 levels are increased in the epidermis of patients with AD [8]. The administration of anti-IL-17A inhibitors in patients with AD resulted in no significant improvement compared with those receiving a placebo for both clinical assessments and lesional skin immunohistochemical analysis [9]. In contrast, although there have been some reports on the mechanism of action of IL-25, the dynamics of IL-25 production are yet to be thoroughly investigated. Therefore, in this study, we assessed IL-25 trends that may influence immune dynamics. In addition, we discuss the implications of this treatment for prolonged AD.

## 2. Results

### 2.1. Cytokine Production by T Helper Cells

2-, 4-, and 6-month-old female transgenic mice, in which the human caspase-1 gene was specifically overexpressed under the keratinocyte-specific keratin 14 promoter (KCASP1Tg) [10,11], and C57BL/6N wild-type littermate (WT) mice were used. The mononuclear cells were purified from the cervical lymph nodes of these mice, and type 1, 2, and 3 cytokine expression levels in CD4 T, CD8 T, and γδ-T cells were measured by flow cytometry. Figure 1A demonstrates the production of IL-4 and IL-13, TNFα and IFNγ, and IL-17A and IL-17F by CD4 T, CD8 T, and γδ-T cells in 4-month-old KCASP1Tg mice. The number of cytokine-producing T cells increased with inflammation over time (Figure 1B,C). Abundant IL-13 and low levels of IL-4 were produced mainly by CD4-positive T cells. TNFα and IFNγ levels increased constantly, especially in CD8 + T cells (Figure 1B,C). Among the type 3 cytokines, IL-17-producing CD4 T cells were subdivided into 17A+/F−, A+/F+, and A−/F+ populations; however, γδ-T cells did not have IL-17A−/F+ population. At 4 months upon the peak of skin eruption in KCASP1Tg mice, the total number of cytokine-producing cells was elated, and at 6 months, total lymphocyte counts decreased with a similar cytokine profile.

### 2.2. Cytokine Production by Innate Lymphoid Cells (ILCs)

The cultured mononuclear cells were gated on ILCs by excluding the cell population stained for surface-specific markers, and cytokine expression levels in ILCs were measured by flow cytometry. Representative figures for type 2 cytokine, IL-4 and IL-13 production, type 1 cytokine, TNFα and IFNγ production, and type 3 cytokine IL-17A and IL-17F production from ILCs are shown (Figure 2A). Similar to T helper cells, IL-13-producing cells were more abundant than IL-4-producing cells, although both IL-4- and IL-13-producing cells were also detected. Cells producing type 1 cytokines were observed in plenty. IL-17A+/F− and A+/F+ populations were detected, but the A−/F+ population was low in the ILC3 population. The total number of ILCs peaked at 4 months and decreased at 6 months of age (Figure 2B,C).

### 2.3. Expression of Nuclear Transcription Factors

mRNA was purified from cultured mononuclear cells before and after 4 h of culture, and the levels of the nuclear transcription factors *T-bet*, *GATA3*, *RoRa*, and *RoRc* were measured by RT-PCR. In KCASP1Tg mice, with prolonged inflammation, the expression levels of *T-bet*, *GATA3*, *RoRa*, and *RoRc* were enhanced in the unstimulated condition (Figure 3). This could be a result of constitutive activation and hence, as expected, the expression levels of these transcription factors were suppressed upon simulation with phorbol myristate acetate and ionomycin.

### 2.4. IL-25 Production by Mononuclear Cells

The cultured mononuclear cells were gated on CD4 T, CD8 T, and γδ-T cells as well as ILCs by excluding the cell population stained for surface markers. In the remainder of the cell population, IL-25 levels were measured by flow cytometry. A representative figure for IL-17E production of these cells is shown in Figure 4A. We observed that the CD8+ T cell population produces ample IL-25. At 6 months of age, IL-25-producing cells increased in a time-dependent manner, even when the acute phase of the skin rash had abated (Figure 4B).

### 2.5. IL-25 Production by Type 2 and 3 Cells

Through intracellular staining of type 2 and type 3 cytokines, we observed that there is an overlap between IL-4- and IL-13-producing cells (Figure 5). However, IL-25-producing cells were distinct from IL-4- and IL-13-producing cells. IL-17F-producing cells simultaneously produce IL-25, and it is not produced from IL-17A+ cells (Figure 5).

## 3. Discussion

The skin epithelium is open to the outside environment. It plays a vital role in immune responses by producing various cytokines and chemokines and serves as a defense line to prevent the entry of different environmental factors. Epithelial cells are of increasing interest in allergic diseases because they may be the starting point of allergic inflammation. Cutaneous inflammation is not only a localized problem of the skin but also causes inflammation of organs throughout the body, resulting in comorbidities and a shortened life expectancy. Overproduction of skin-derived inflammatory cytokines results in organ failures, such as cardiovascular and cerebrovascular disorders [12,13] and systemic amyloidosis [13,14,15,16,17]. Many cytokines released from active skin lesions of persistent inflammation may be mixed in the bloodstream, causing the liver to produce amyloid A protein. Sometimes, it leads to inflammation of the adipose tissue, resulting in the secretion of adipocytokines. Later, the activated mononuclear cells infiltrate the fatty tissue, which may influence the surrounding tissue directly, including the significant artery [18]. The concept of hardening of this vasculature is known as “the inflammatory skin march” [19]. Inflammatory skin diseases may also complicate osteoporosis by decreasing the vascular network in the bone marrow, increasing the number of osteoclasts and decreasing osteoblasts [20]. Hypoplasia and decreased sperm motility are causes of male infertility, and increased production of inflammatory cytokines due to skin lesions is presumed to be the direct cause [21]. In addition, degeneration of the salivary glands and decreased saliva production occur upon sustained inflammation [22].

Although AD is generally regarded as a type 2 disease, no reports have described the quantitative and qualitative dynamics of cytokines in AD. Our analysis of helper T cells and ILCs showed that type 1 immunity, which is most active in biological defense, was also predominant in AD mice. This immune response peaked in the acute phase at 4 months of age and decreased later on in the chronic phase at 6 months of age. Previous studies have suggested that the spleen, which is involved in lymphocyte maturation, may be immunosuppressive in the chronic phase owing to the destruction of the lymph follicle architecture caused by amyloid deposition, possibly leading to decreased lymphocyte biosynthesis [13,16]. In helper T cells, the number of type 2 and type 3 cytokine-producing cells was higher than that of ILCs, suggesting a greater diversity of selection. Although there have been speculations regarding the balance between cells producing type 1, type 2, and type 3 cytokines being equivalent [23], this needs to be further explored. Reports on the adverse effects of cancer therapy with immune checkpoint inhibitors have demonstrated the importance of regulatory/suppressive cells. Our qualitative and quantitative analyses proved that type 1 cells are more abundant than type 2 and type 3 cells, which makes type 1 immunity a dominant player in AD (Figure 6). Moreover, the fragile balance between type 1 and type 2 was revealed. This is supported by the fact that, in clinical practice, the use of IL-17 inhibitors in the treatment of psoriasis can lead to AD and, conversely, the use of IL-4/13 inhibitors in the treatment of AD can lead to psoriasis-like skin rashes. Furthermore, upon prolonged inflammation, we reported that the *T-bet*, *GATA3*, and type 3 transcription factors *RoRa* and *RoRc* were inherently enhanced; however, this was suppressed upon simulation, probably because the immune system was exhausted.

The IL-17 family of cytokines is key to host defense responses and inflammatory diseases. Among the IL-17 family, IL-25 (IL-17E) was identified as a novel IL-17 cytokine family produced by Th2 cells [24]. Subsequently, mast cells, basophils, eosinophils, NKT cells, macrophages, ILC2, Th9 cells, and epithelial cells have also been reported to produce IL-25 [25,26,27]. Compared with other IL-17 cytokine family members, IL-25 has a relatively low similarity to IL-17A and exhibits a distinct function from other IL-17 cytokines. IL-25 binds to its receptor composed of IL-17 receptor A (IL-17RA) and IL-17 receptor B (IL-17RB) for signal transduction. IL-25 has been implicated as a type 2 cytokine that can induce the production of IL-4, IL-5, and IL-13 [24], which in turn inhibits the differentiation of T helper (Th) 17 cells.

IL-25 promotes neutrophil recruitment by activating primary human macrophages. IL-25 is highly expressed in skin lesions of patients with psoriasis and AD [28]. Furthermore, IL-25 is involved in disease exacerbation by decreasing filaggrin expression, a gene related to skin barrier function [26,29]. On the contrary, the combined blockade of type 2 cytokines, namely, IL-13 and IL-25, was more effective than the obstruction of either cytokine in reducing the infiltration of inflammatory cells in the airways with attenuated airway hyperresponsiveness and tissue remodeling [30]. Hence, the IL-25 embargo may represent a promising strategy for targeting skin inflammation [31].

The current investigation showed that IL-25 production was higher in CD8 T cells than in CD4 T cells, γδ T cells, and ILCs (Figure 4). Although IL-25 is a type 2 cytokine, IL-25-producing cells are distinct from IL-4- and IL-13-producing cells. In addition, we showed that IL-25 is concurrently produced by IL-17F but not IL-17A-producing cells. The number of IL-25-producing cells increased in a time-dependent manner. Figure 1 and Figure 2 show that cells in the inflow lymph nodes and cytokine-producing cells increased during the acute phase of skin lesions at 4 months and decreased during the chronic phase at 6 months of age. However, IL-25 differs from IL-17, IL-4, and IL-13 in its oscillatory pattern, and shows a further increase during the chronic phase, which may contribute to the prolongation of type 2 inflammation. This finding suggests a possible treatment strategy for IL-25 inhibition during the chronic phase of inflammation.

In conclusion, we reported that, among the cytokine responses that occur during allergic dermatitis, type 1 cells are dominant at all times, whereas type 2 and type 3 cytokine-producing cells are equally present. We also found that IL-25-producing cells, which belong to the IL-17 family and overlap with IL-17F-producing cells but have the predisposition to type 2, are persistently increased upon prolonged inflammation. A limitation of this study is that only one strain of mouse was used, and there is a need to follow up with other mouse models of AD and eventually with human studies. However, these results also suggest that anti-IL-25 antibodies may be of prime importance to treat AD in the future.

## 4. Materials and Methods

### 4.1. Atopic Dermatitis Model Mice KCASP1Tg Used in the Study

Two-, 4-, and 6-month-old female transgenic mice, in which the human caspase-1 gene was expressed under the KCASP1Tg [3,10,11], were used as an AD model, and C57BL/6N littermate (WT) mice were used as controls. The mice were housed in an environmentally conditioned room at 21 ± 2 °C with a 12/12 h light cycle, 60% humidity, and food and water available ad libitum. Animal care was conducted according to the current ethical guidelines, and the Mie University Board Committee approved the experimental protocol for Animal Care and Use (#22-39-5-1).

### 4.2. Intracellular Cytokine Staining in T Helper Lymphocytes

Cervical lymph nodes were sampled from 2-, 4-, and 6-month-old KCASP1Tg and WT mice. After washing, the cells were filtered through a mesh and incubated with ACK Lysing Buffer (Thermo Fisher Scientific, Waltham, MA, USA) to lyse the erythrocytes. Mononuclear cells were isolated and purified using density gradient centrifugation. They were then cultured in RPMI media supplemented with 50 ng/mL of phorbol myristate acetate (AdipoGen, San Diego, CA, USA) along with 1 µg/mL of ionomycin (Fujifilm, Tokyo, Japan) for 4 h. The LIVE/DEAD™ Fixable Aqua Dead Cell Stain Kit (Thermo Fisher Scientific) was used to exclude apoptotic and necrotic cells. The cultured mononuclear cells were stained with surface antibodies: CD4-Brilliant Violet 421, CD8-APC-Cy7, and γδ-TCR-APC (BD Biosciences, Franklin Lakes, NJ, USA) in cell surface staining buffer containing 0.1 M phosphate-buffered saline and 2% FCS (Biowest, Nuaillé, France) and then stained with IFNγ-Brilliant Violet 605 and TNFα-FITC, IL-4- Brilliant Violet 711, IL-13-PE, IL-17A-Brilliant Violet 786, and IL-17F-PerCp-Cy5.5 antibodies (BD Biosciences). The expression patterns of inflammatory cytokines were analyzed using a BD Lyric flow cytometer (BD Biosciences).

### 4.3. Intracellular Cytokine Staining in ILCs

The lineage markers (PerCp-Cy5.5-CD3, CD4, CD8a, CD45R, Gr-1, RceRI, and Siglec-F, Biolegend, San Diego, CA, USA) were used to classify ILCs and then stained with IFN-γ-Brilliant Violet 605, TNFα-APC, IL-4- Brilliant Violet 421, IL-13-FITC, IL-17A- APC-Cy7, and IL-17F-PE antibodies (BD Biosciences).

### 4.4. Real-Time Polymerase Chain Reaction (PCR) Analysis for the Expression of Nuclear Transcription Factors

Total RNA was extracted from 4-month-old mice using Tri Reagent (Molecular Research Center, Cincinnati, OH, USA). The RNA concentration was measured using a NanoDrop Lite spectrophotometer (Thermo Fisher Scientific). One microgram of total RNA was used to synthesize cDNA using a High-Capacity RNA-to-cDNA Kit (Applied Biosystems, Foster City, CA, USA). TaqMan Universal PCR Master Mix II with UNG (Applied Biosystems) was used to measure the mRNA expression of the T-box master transcription factor regulator (*Tbx21*, Mm00450960_m1), GATA-binding protein-3 (*Gata3*, Mm00484683_m1), retinoic acid-related orphan receptor A (*RoRa*, Mm01296312_m1), and RoRc (*RoRc*, Mm01261022_m1). Glyceraldehyde-3-phosphate dehydrogenase (*GAPDH* Mm99999915_g1) was used as the internal control. All probes were purchased from Applied Biosystems and amplification was performed using a LightCycler 96 System (Roche Diagnostics, Indianapolis, IN, USA). The cycling parameters were as follows: 50 °C for 120 s and 95 °C for 600 s, followed by 40 cycles of amplification at 95 °C for 15 s and 60 °C for 60 s.

### 4.5. IL-25 Cytokine Production from T Helper Lymphocytes and ILCs

The cultured mononuclear cells were stained with surface antibodies: CD4-Brilliant Violet 421, CD8-APC-Cy7, γδ-TCR-FITC, and the ILCs’ lineage markers (PerCp-Cy5.5-CD3, CD4, CD8a, CD45R, Gr-1, RceRI, and Siglec-F) in cell surface staining buffer containing 0.1 M phosphate-buffered saline and 2% FCS (Biowest, Nuaillé, France), followed by staining with IL-25-Alexa647 antibodies (R&D Systems, Minneapolis, MI, USA).

### 4.6. Statistical Analysis

Statistical analyses were performed using PRISM software version 9 (GraphPad, San Diego, CA, USA). Ordinary one-way ANOVA followed by Tukey’s multiple comparison test was used to compare data between groups. n.s., not significant; *, *p* < 0.05; **, *p* < 0.01; ***, *p* < 0.001; ****, *p* < 0.0001.

## Figures and Tables

**Figure 1 ijms-24-03310-f001:**
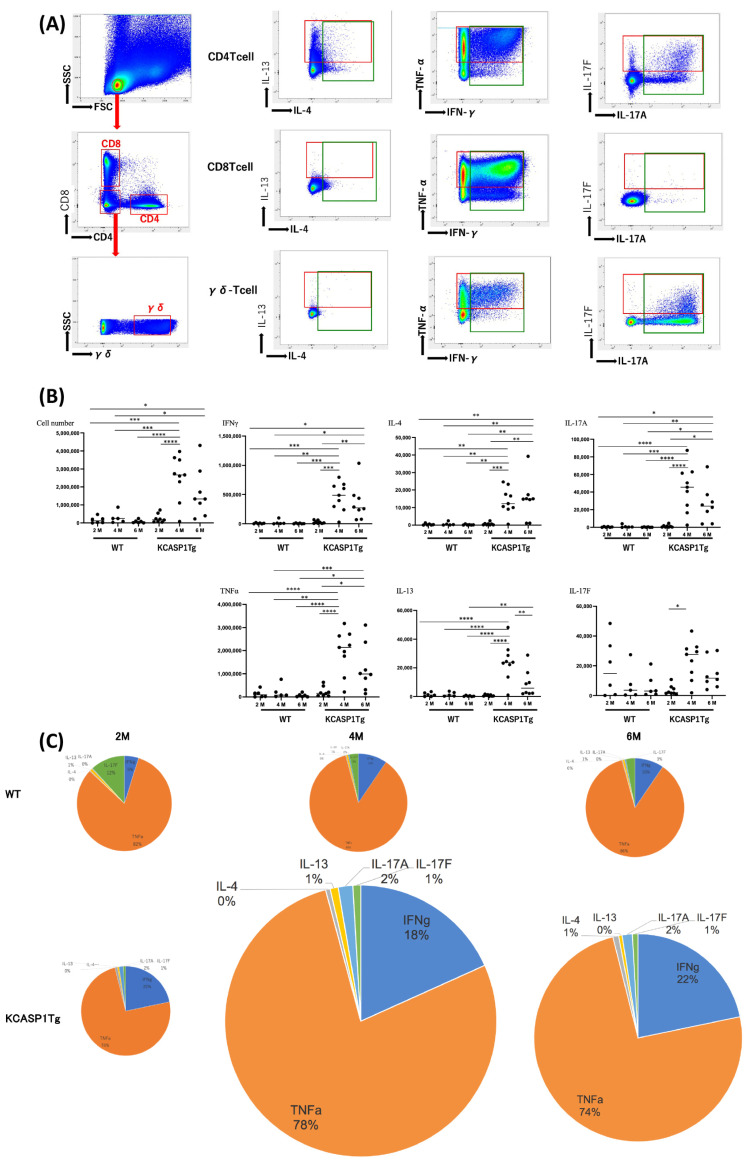
Cytokine production by T helper cells. (**A**) The plots represent the production of specified cytokines by the three types of T cells namely, CD4 T, CD8 T, and γδ-T cells, in 4-month-old KCASP1Tg mice. CD4T cells produce IL-13 and a small amount of IL-4; TNFα-producing cells are predominant for type 1 cytokines, especially from CD8T cells and CD4T cells; CD4T cells are divided into IL-17A alone, IL-17F alone, and IL-17A/F-producing cells. On the other hand, γδ T cells do not produce IL-17F alone. (**B**,**C**) The number of indicated cytokine-producing T cells was measured at 2, 4, and 6 months of age (M). All cytokine-producing cells were high in number at the peak of the dermatitis at 4 months of age, with a decreasing trend in cell number at 6 months of age when the skin rash became chronic. (**C**) Pie chart of the data shown in (**B**). The size of the graph in (**C**) reflects the total number of T helper cells. Asterisks indicate a significance difference based on a one-way ANOVA and Tukey’s multiple comparison test (*, *p* < 0.05; **, *p* < 0.01; ***, *p* < 0.001; ****, *p* < 0.0001).

**Figure 2 ijms-24-03310-f002:**
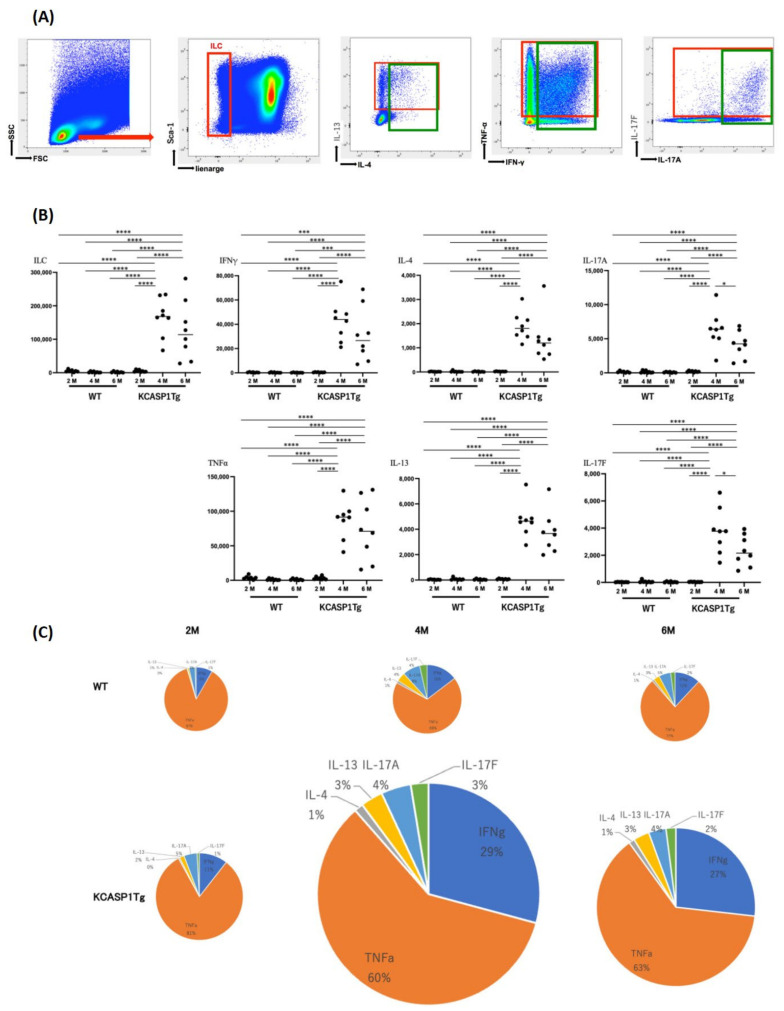
Cytokine production by ILCs. (**A**) The plots represent type 1, 2, and 3 cytokine production by ILCs in 4-month-old KCASP1Tg mice as observed by flow cytometry. (**B**,**C**) The number of indicated cytokine-producing ILCs was measured at 2, 4, and 6 months of age (M). (**C**) Pie chart of the data shown in (**B**). The total number of ILCs peaked at 4 months and decreased at 6 months. The size of the graph in (**C**) is representative of the number of ILCs. Asterisks indicate significance difference based on a one-way ANOVA and Tukey’s multiple comparison test (*, *p* < 0.05; ***, *p* < 0.001; ****, *p* < 0.0001).

**Figure 3 ijms-24-03310-f003:**
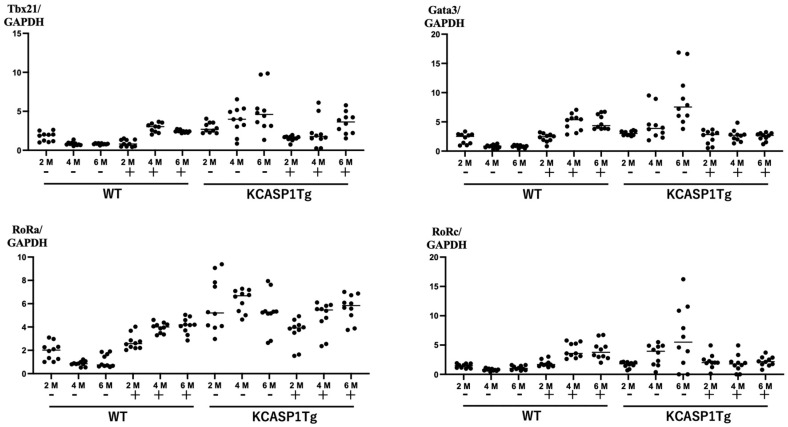
Expression of nuclear transcription factors. The graphs represent the relative expression of the indicated transcription factors with respect to the housing-keeping gene-GAPDH of mononuclear cells before and 4 h post-culture. In KCASP1Tg, with prolonged inflammation, *T-bet*, *GATA3*, *RoRa*, and *RoRc* transcription factors were enhanced in the unstimulated condition, probably owing to the constitutive activation, which was suppressed upon simulation.

**Figure 4 ijms-24-03310-f004:**
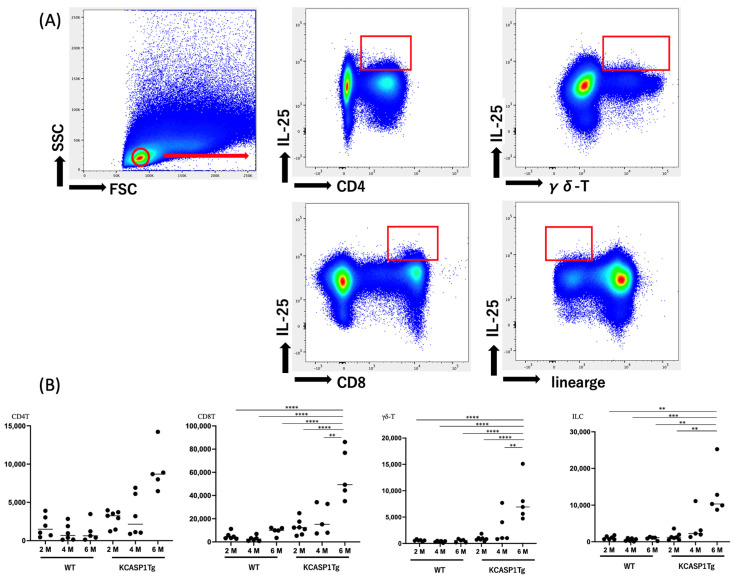
IL-25 production by mononuclear cells. (**A**) Focusing on the lymphocyte fraction of cultured cells, we detected CD4T cells, CD8T cells, γδ-T cells, and cells producing IL-25 from ILCs. The plots represent the production of IL-25 by indicated cell types as observed by flow cytometry. (**B**) The number of indicated cytokine-producing cells was measured at 2, 4, and 6 months of age (M). IL-25-producing cells increased in a time-dependent manner even at 6 months of age. Asterisks indicate significance difference based on a one-way ANOVA and Tukey’s multiple comparison test (**, *p* < 0.01; ***, *p* < 0.001; ****, *p* < 0.0001).

**Figure 5 ijms-24-03310-f005:**
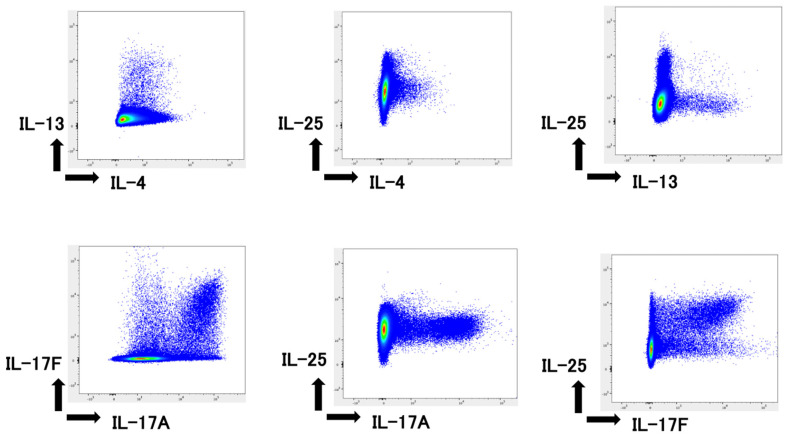
IL-25 production by type 2 and type 3 cells. The plots represent the co-production of indicated type 2 and type 3 cytokines as observed by flow cytometry. Intracellular staining results for type 2 and type 3 show some overlap between IL-4- and IL-13-producing cells. However, IL-25-producing cells also proved to be distinct from IL-4- and IL-13-producing cells. As for type 3 cytokines, IL-25 might be simultaneously produced by IL-17F-producing cells and not by IL-17A-producing cells.

**Figure 6 ijms-24-03310-f006:**
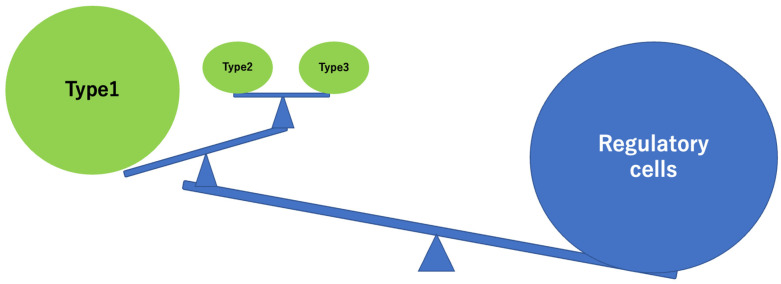
The immunological balance between four factors. Based on our analyses, this cartoon represents the balance between type 1, type 2, and type 3 immunity with respect to the regulatory cells in AD. The size of the rounds represents the abundance in the indicated cell types.

## Data Availability

Not applicable.

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
