# Peer review of "The Interplay of Type 1, Type 2, and Type 3 Lymphocytes and Cytokines in Atopic Dermatitis"

_ijms, 2023, doi:10.3390/ijms24043310_

Round 1

Reviewer 1 Report

The authors analyzed the sequential changes of Type 1, 2, and 3 inflammatory cytokines in lymphocytes purified from cervical lymph nodes using a mouse model of atopic dermatitis in which caspase-1 was specifically amplified under the keratin induction. The cells were cultured and stained for CD4, CD8, and γδTCR, and then intracellular cytokines. The nuclear transcription factor was calculated using RT-PCR. The cytokine-producing T cells increased with inflammation over time. The total number of T cells and ILCs peaked at four months and decreased at the chronic phase. The authors found that the IL-25-producing cells proved distinct from IL-4- and IL-13-producing cells and concluded that the IL-25 inhibition during chronic inflammation may be an effective therapy.

This study was very well-written. Could the authors please provide limitations and strengths and the future direction of this study?

Author Response

Responses to the comments of Reviewer #1

Comments to the Author:

  1. The authors analyzed the sequential changes of Type 1, 2, and 3 inflammatory cytokines in lymphocytes purified from cervical lymph nodes using a mouse model of atopic dermatitis in which caspase-1 was specifically amplified under the keratin induction. The cells were cultured and stained for CD4, CD8, and γδTCR, and then intracellular cytokines. The nuclear transcription factor was calculated using RT-PCR. The cytokine-producing T cells increased with inflammation over time. The total number of T cells and ILCs peaked at four months and decreased at the chronic phase. The authors found that the IL-25-producing cells proved distinct from IL-4- and IL-13-producing cells and concluded that the IL-25 inhibition during chronic inflammation may be an effective therapy.

This study was very well-written. Could the authors please provide limitations and strengths and the future direction of this study?

Response: Thank you for your suggestions. The cytokine balance that occurs during allergic dermatitis was found to be overwhelmingly Type 1 dominant at all phases, with Type 2 and Type 3 cytokine-producing cells maintaining an equal degree of balance. We also found that IL-25-producing cells, which belong to the IL-17 family but have a predisposition to Type 2, are persistently increased as inflammation becomes more prolonged. This event may contribute to the prolongation of Type 2 inflammation. A limitation of this study is that only one strain of mice was used, and there is a need to follow up with other mouse models of atopic dermatitis and eventually with human studies. However, the results also suggest that anti-IL-25 antibodies may be useful in the future.

We have supplemented the limitations, strengths, and future direction in the discussion section. We appreciate your comment.

Reviewer 2 Report

Comments for authors

The manuscript entitled “The sequential dynamics of Type 1, Type 2, and Type 3 lym-phocytes in the atopic dermatitis model mice and the association with IL-17E” is although interesting work, but there is many limitations which decrease the interest of the work.

The manuscript is written in a very poor manner. It needs substantial improvement.

1. In abstract please rephrase the sentence “This suggests that IL-25 inhibition during the chronic inflammation may be the effective therapy” with “The findings suggest that the inhibition of IL-25 may be a potential target in the treatment of inflammation”.

2. The introduction is too short, please rewrite the introduction with clear theme and correlations of the research project.

3. In material and methods please revise the headings thoroughly, as first subsection in this section is “mouse study” what is this?? Please write the complete model’s name or with experimental animals.

4. Where is the conclusion of the study?

The manuscript should must be improved.

Author Response

Responses to the comments of Reviewer #2

Comments to the Author:

The manuscript entitled “The sequential dynamics of Type 1, Type 2, and Type 3 lym-phocytes in the atopic dermatitis model mice and the association with IL-17E” is although interesting work, but there is many limitations which decrease the interest of the work.

  1. The manuscript is written in a very poor manner. It needs substantial improvement.

Response: This paper was proofread by a native English-speaking scientist.

  1. In abstract please rephrase the sentence “This suggests that IL-25 inhibition during the chronic inflammation may be the effective therapy” with “The findings suggest that the inhibition of IL-25 may be a potential target in the treatment of inflammation”.

Response: Thank you for your suggestions. We have changed the sentence, as the reviewer pointed out.

  1. The introduction is too short, please rewrite the introduction with clear theme and correlations of the research project.

Response: Thank you for your suggestions. We have supplemented the information in the introduction session.

  1. In material and methods please revise the headings thoroughly, as first subsection in this section is “mouse study” what is this?? Please write the complete model’s name or with experimental animals.

Response: We have modified the material and methods section thoroughly.

  1. Where is the conclusion of the study? The manuscript should must be improved.

Response: Thank you for your suggestions. We have supplemented the following comments.

The cytokine responses that occur during allergic dermatitis, type 1 cells are dominant at all times, whereas, type 2 and type 3 cytokine-producing cells are present equally. We also found that IL-25-producing cells, which belong to the IL-17 family and overlap with IL-17F-producing cells but have the predisposition to type 2, are persistently increased upon prolonged inflammation. A limitation of this study is that only one strain of mouse was used, and there is a need to follow up with other mouse models of AD and eventually with human studies. However, these results also suggest that anti-IL-25 antibodies may be of prime importance to treat AD in the future.

Round 2

Reviewer 2 Report

Dear Editor,

Thank you for giving me the opportunity to review the revised manuscript.

Manuscript can be accepted for publication, but quality of all the figures must be improved.

Thank you.

Regards,

Abdul Basit

Author Response

Responses to the comments of Reviewer #2

Comments to the Author:

Thank you for giving me the opportunity to review the revised manuscript.

Manuscript can be accepted for publication, but quality of all the figures must be improved.

Thank you.

Regards,

Abdul Basit

Response:

Dear Prof. Abdul Basit,

Thank you for reviewing our project. After consulting with the assistant editor, we decided to put together a diagram of each. I think this makes it easier to see. We appreciate your help.